# Mode of Action of Brassinosteroids: Seed Germination and Seedling Growth and Development—One Hypothesis

**DOI:** 10.3390/ijms26062559

**Published:** 2025-03-12

**Authors:** Bogdan Nikolić, Vladan Jovanović, Branislav Knežević, Zoran Nikolić, Maja Babović-Đorđević

**Affiliations:** 1Institute for Plant Protection And Environment, Teodora Drajzera Str., No. 9, 11040 Belgrade, Serbia; 2Institute for Pesticides and Environmental Protection, Banatska Str., No. 31b, 11080 Belgrade, Serbia; vladanjo11@gmail.com; 3Department of Crop and Vegetable Sciences, Faculty of Agriculture, University of Priština in Kosovska Mitrovica, 38219 Lešak, Serbia; branislav.knezevic@pr.ac.rs; 4Department for Fruit Growing and Viticulture Sciences, Faculty of Agriculture, Univerzity of Priština in Kosovska Mitrovica, 38219 Lešak, Serbia; zoran.nikolic@pr.ac.rs; 5Department of Plant Protection, Faculty of Agriculture, University of Priština in Kosovska Mitrovica, 38219 Lešak, Serbia; maja.babovic@pr.ac.rs

**Keywords:** brassinosteroids, 24-epibrassinolide, mode of action, source–sink relationship, plant seedling energetics, metabolism, growth and development

## Abstract

Brassinosteroids, as unique plant steroid hormones that bear structural similarity to animal steroids, play a crucial role in modulating plant growth and development. These hormones have a positive impact on plant resistance and, under stressful conditions, stimulate photosynthesis and antioxidative systems (enzymatic and non-enzymatic), leading to a reduced impact of environmental cues on plant metabolism and growth. Although these plant hormones have been studied for several decades, most studies analyze the primary site of action of the brassinosteroid phytohormone, with a special emphasis on the activation of various genes (mainly nuclear) through different signaling processes that influence plant metabolism, growth, and development. This review explores another issue, the secondary influence (the so-called mode of action) of brassinosteroids on changes in growth, development, and chemical composition, as well as thermodynamic and energetic changes, mainly during the early growth of corn seedlings. The interactions of brassinosteroids with other phytohormones and physiologically active substances and the influence of these interactions on the mode of action of brassinosteroid phytohormones were also discussed. Seen from a cybernetic point of view, the approach can be labeled as “black box” or “gray box”. “Black box” and “gray box” are terms for cybernetic systems, for which we know the inputs and outputs (in an energetic, biochemical, kinetic, informational, or some other sense), but whose internal structure and/or organization are completely or partially unknown to us. The findings of many researchers have indicated an important role of reactive species, such as oxygen and nitrogen reactive species, in these processes. This ultimately results in the redistribution of matter and energy from source organs to sink organs, with a decrease in Gibbs free energy from the source to sink organs. This quantitative evidence speaks of the exothermic nature and spontaneity of early (corn) seedling development and growth under the influence of 24-epibrassinolide. Based on these findings and a review of the literature on the mode of action of brassinosteroids, a hypothesis was put forward about the secondary effects of BRs on germination and the early growth of plant seedlings.

## 1. Introduction

Seed germination and seedling establishment are among the key developmental stages of plants, and are very sensitive to environmental variations [1]. As a process of transition from a state of anabiosis and non-differentiation to a state of full metabolic and developmental activity, germination is accompanied by energy changes [2]. During the processes, the reserve substances (starch and storage proteins) are degraded via a series of intermediate compounds (saccharides and peptides) to monosaccharides and amino acids, which serve as substrates for respiration and biosynthesis in the development of young plants [3,4].

This process is of particular interest regarding grass seeds, both because of their great economic importance and because they represent model systems, for germination itself as well as for studying the effects of various endogenous and exogenous factors on the germination process [5]. Most important endogenous factors affecting the germination process (and the anabiosis that preceded it) in grass seeds include the actions of phytohormones: gibberellins (GAs) and abscisic acid (ABA) [5,6,7,8]. Other endogenous (phytohormones, sugars) and exogenous (oxygen) factors can modify the influence of GAs and ABA on seed germination [6,9,10,11,12,13]. During anabiosis and in the first stages of germination, seeds and seedlings are energetically and cybernetically metastable, even unstable systems, and their structures and functions can easily be disrupted, especially due to the production of reactive species during oxidative stress. [1,2,14,15]. Therefore, it is important to determine the factors that contribute to overcoming that “critical state” in the development of young plants, during so-called seedling establishment. Recent results from other biological disciplines suggest that there is a tendency to maintain the stability of these metastable structures of living matter through controlled modulation of the energy of a system through entropy [16], which may be of interest when considering seed germination and the initial stages of seedling growth, a cybernetic point of view, the approach can be labeled as “black box” or “gray box” [17].

## 2. Observations Regarding the Applied Research Methodology

Mode of action as a technical term was introduced by Moreland [18] in the field of pesticide science (especially herbicides), meaning a differentiation between the primary action of some chemicals on plant metabolism (mechanism of action) and the subsequent physiological, developmental, growth, and other effects of these compounds (mode of action), either in terms of self-regulation of plant metabolism in the presence of xenochemicals or the direct impact of xenochemicals on plant metabolism. As many important agronomic traits of crops (biological vigor of seedlings, crop resistance to stress, crop yield) are polygenic traits, these traits are most easily considered in relation to the mode of action of chemicals, in this case brassinosteroid (BR) phytohormones.

Regarding the mode of action of BR phytohormone, an overview of multi-annual tests of exogenous treatments of seedlings of two corn hybrids (ZP434 and ZP704; differing in drought stress resistance and yield, according to data from the producer of these corn hybrids (Maize Research Institute, Belgrade, Serbia) based on many years of field trials and publications by associates of this institution) describes different aspects of the mode of action of 24-epibrassinolide (24-EBL) applied to corn seedlings in a wide range of concentrations (from 5.2 × 10^−15^ to 5.2 × 10^−7^ mol of 24-EBL). We treated seedlings of two maize hybrids with the aforementioned concentrations of 24-EBL. We germinated 50 seeds (in four replicates) of the two used corn hybrids using the standard ISTA (International Seed Testing Association) method, which involved soaking the corn seeds and wrapping them in filter papers soaked in the appropriate concentrations of 24-EBL. The concentrations of 24-EBL with which we treated seedlings of the two mentioned maize hybrids, 7 days old, were selected based on data from the literature, starting from inhibitory concentrations (5.2 × 10^−7^ mol of 24-EBL) up to concentrations that are claimed in the literature to cause only homeopathic effects (5.2 × 10^−15^ mol of 24-EBL); we also tested the effect of the aforementioned concentrations of 24-EBL in field experiments, as shown in work [19]. Assuming that BR of the same concentrations affects plant metabolism as well as their biosynthesis [20], and due to the small size of corn seedlings as a plant model, we believe that the chosen methodological approach corresponds to the natural processes during corn seed germination and the first phases of the growth and development of seedlings of these plants, representing the grass family as a very significant crop.

The obtained results were compared with the findings of many other researchers, obtained on the same or similar plant systems, and the similarities and/or differences between the two types of findings were discussed.

The methodology used in our research and the research of other authors is described in the references mentioned, in illustrations and captions, as well as in the Appendix A.

## 3. Germination of Seeds of Different Plant Species—Early Phases of Seedling Growth and Development and Some Processes Affecting Seedling Growth and Development Under the Influence of Different Brassinosteroids

The Appendix A present many parameters related to germination and the early growth of different plants, as influenced by various concentrations and types of brassinosteroid (BR) phytohormones (Appendix A) [21,22,23,24,25,26,27,28,29,30,31,32,33,34,35,36,37,38]. Different concentrations and types of BRs are shown to have a variety of effects on germination, the mass redistribution of seedling parts, and other growth parameters of young plants of different species. For example, Appendix A shows that a wide range of concentrations (from 5.2 × 10^−12^ M to 10^−6^ M) of two types (24-EBL and 28-HBL: 28-homobrassinolide) of BR phytohormones affect the process of seed germination in different plant species (corn, *A. thaliana*, *Brasicca juncea*, *Raphanus sativus*).

Likewise, focusing on the changes in various growth parameters observed (Appendix A) in the seedlings of some plant species (corn, rice, soybean, *A. thaliana*, *Wolffia arrhiza*, *Vigna radiata*, *Brasicca juncea*, *Raphanus sativus*), it was evident that they were mainly positively influenced by the wide range of concentrations (from 5.2 × 10^−15^ M to 10^−6^ M) of several types of brassinosteroids (brassinolide (BL), CS (castasterone), 24-EBL, and 28-HBL), either in association with other environmental influences (drought stress, extreme temperatures, heavy metals, etc.) or without them.

However, when we compare % germination (Appendix A), as a physiological–morphological indicator of the action of various forms of BR phytohormones, with other physiological–morphological indicators of phytohormone action (Appendix A), its discriminativeness, especially with a smaller BR phytohormone concentration, is significantly lower. Therefore, we believe that it is necessary to use more morpho-physiological parameters and a larger range of applied BR phytohormone concentrations in order to obtain an appropriate discriminative response for the modes of action of brassinosteroid phytohormones.

Also, in some of the works mentioned in Appendix A, the same parameters were analyzed in different parts of the seedlings (P: plumule/shoot, R: radicle/root, RoS/S: rest of seed), and it was noted that the concentration of BRs needed to cause an effect higher or lower than the control was different in different organs of seedlings of particular species/plant genotypes. This is reminiscent of the findings of Van Esse et al. [39] that brassinosteroid receptors (BRIs) are not evenly distributed in various tissues of young plants. What kind of processes underlie all these different phenomena? (Figure 1).

Figure 1a,b indicate similarity in the effects of different concentrations of exogenous 24-EBL on the seedling growth parameter vigor index II and their influence on changes in the corn seedling content of ROS and RNS (oxygen and nitrogen) radical species, which are significant in numerous developmental and plant growth processes, such as the remodeling of plant cell walls [41,42], cell division, and the growth of young plants [43]. Equivalent measurements of redox capacity (as a measure of the content of ROS and RNS species) in corn seedlings were carried out in a trial presented by Božilović et al. [21] (Appendix A).

## 4. Some Chemical Changes During Seed Germination and the Early Development and Growth of Seedlings of Different Plant Species Under the Influence of Different Brassinosteroids

In Appendix A [21,22,23,24,25,26,27,28,29,30,31,32,33,34,35,36,37,38], we may note that heavy metals act on their own, and their effect is also modified by brassinosteroids, which prompted us to study the interaction of BRs and mineral elements during seed germination and in the early stages of the growth and development of seedlings of various species in depth in the following metadata review (Appendix A) [22,44,45].

Appendix A shows that Waisi et al. [22,44,45] observed that a wide range of concentrations of 24-EBL differentially affected the content of various elements in different organs of seedlings of two corn genotypes (ZP434, ZP704), which agrees with the findings of van Esse et al. [40] on the differences in the abundance of receptors for BRs in various plant organs. It appears that the different degrees of sensitivity to BRs (due to different abundances of receptors for brassinosteroids) in various organs of corn seedlings could lead to the redistribution of the macroelements phosphorus and potassium, oligoelements (Na, Mg), as well as some microelements and heavy metals in different organs of corn seedlings of different genotypes. Of course, this is only a working hypothesis that should be verified by other methods.

The present overview (Appendix A) highlights the works of other authors who have examined the interactions of BRs and chemical elements (mainly heavy metals) under conditions that either simulate or actually pose real environmental stress. Cao et al. [37] examined the interactions of BRs and Cd together with various protective substances (GSH, GB, SA) and their effect on the content of several microelements (Fe, Mn, Zn, Cu) in the plumule and radicle of rice seedlings, and found that the mentioned manipulative treatments increased the content of iron but decreased the content of other microelements, indicating their different roles in the early stages of rice development. Many studies conducted by Indian researchers [32,34,46] examined the effect of several different BRs phytohormones (CS, 24-EBL, 28-HBL) on the content and bioaccumulation (BCF) of some microelements (Cu, Mn, Zn, Co) and other metals (Ni) in plants. The researchers found that the mentioned BRs reduced the content of these elements in the seedlings of *Brassica juncea* plants.

The group of compounds that connects seed germination and the growth of young seedlings with phytohormones is evidently sugars. In addition to their role in energy metabolism and their participation in metabolic transformations, together with organic acids and amino acids [47], it is clear that through polymerization they form various cell structures of young plants, primarily the cell wall [41,48,49]. It is important to mention the signaling role of sugars, especially glucose and sucrose [50], and also their protective role over sensitive cellular structures during stress [51,52]. It is clear that during seed germination and early growth of young plants, complex reserve polymeric compounds (starch, reserve proteins) are broken down, producing simpler compounds (e.g., oligosaccharides, monosaccharides, and amino acids), which serve as substrates for biosynthetic processes in young plants. All these processes are regulated by the relationship between gibberellin and abscisic acid, which are also themselves modulated by brassinosteroids, among other factors [11,12,24]. So, BRs regulate the mutual interactions between abscisic acid and gibberellic acid, and thereby induce the transition from metastable seed dormancy to the dynamic state of seed germination.

The processes are shown in Appendix A, where metadata on brassinosteroid–sugar interactions are assembled. An interesting approach to the study of this problem is its examination in the context of effects of other factors, such as stress from extreme temperatures [29,33] or the addition of analogues, i.e., inhibitors of BR action/synthesis [33,35], but the obtained results are not unequivocal (Appendix A). The situation becomes somewhat clearer when changes in the content of various sugars with different functions in plant metabolism are monitored in seedlings under the influence of BRs. For example, it is interesting to observe changes in sucrose, glucose, and fructose contents in BR-treated seedlings, as they are metabolically related sugars with various energy and signaling functions in plant metabolism. It was observed [44,45] in the majority of treated seedlings of two corn hybrids (ZP434, ZP704) that when the sucrose content was higher than the control values, the values of glucose and fructose, the monosaccharides that form sucrose, were mostly lower than the control values (Appendix A), which means that sucrose synthesis occurs during these processes, and it is possible that these monosaccharides also participate in other metabolic processes, reducing their content in parts of the seedlings of these two corn hybrids. It is also of interest to observe changes in the content of two sugars, arabinose and raffinose, which have a significant role in the construction of plant cell walls in seedlings treated with brassinosteroids, since phytohormones are known to be able to affect cell wall metabolism [42]. Here too, we observe (Appendix A) a certain increase in the content of these sugars under the influence of BRs, whereby different doses of 24-EBL are needed in different organs of corn seedlings to produce the mentioned effects. We assume that the increase in content of these sugars under the influence of BRs occurs as a result of intense cell division during the early stages of the growth and development of seedlings of these two genotypes of corn, which is why these sugars are needed as “building material” [41,48,49]. The protective sugar trehalose also undergoes changes in content (Appendix A) [51,52] in seedlings of the same two hybrids of corn treated with different concentrations of 24-EBL. Differences were detected in the reactions of different organs of the seedlings, as well as across corn genotypes (ZP434, ZP704).

Returning to the initial assumption about compatibility in the early seedling growth and development (Figure 1a) of two corn hybrids treated with various concentrations of 24-EBL and the production of reactive (oxygen and nitrogen) species (Figure 1b) in these seedlings, we come to a problem. Controlling the production of the aforementioned reactive species, i.e., “keeping” them within certain limits [53,54] and modulating their production [53,54], as a way to remodulate the growth process of young plants, is also a problem, since excessive production of the aforementioned reactive species can lead to degradative processes (oxidative shock) and cause the decline of young plants [2,3].

This problem—the removal of reactive (oxygen and nitrogen) species, i.e., antioxidant systems—was addressed in our semiquantitative analysis of the polyphenolic profile of plumule and radicle seedlings of two corn hybrids (ZP434, ZP704) treated with different concentrations of 24-EBL (Appendix A) [55]. Based on these findings, a statistical PCA significance analysis was performed (Appendix A) [55].

Two types of HPTLC methods [53] were used to separate the polyphenols of the plumules and radicles of two corn hybrids (ZP434, ZP704). The first one was found to differentiate polyphenols into three fractions with different data variances (Appendix A) [55], with two data groups being characteristic of different corn seedling hybrids and corresponding to the less polar plumule and radicle polyphenols of the corn seedling hybrids (Appendix A). The variability in the results is more pronounced in hybrid ZP704 (Appendix A) [55]. The influence of the third polyphenolic component is minor. Examinations performed by the other HPTLC method yielded four components of polyphenolic fractions (Appendix A) [55].

The two most abundant components correspond to the polyphenols of the seedlings of the two corn hybrids (Appendix A), while a different grouping of plumule and radicle polyphenols is observed as being especially pronounced in the ZP434 corn hybrid seedlings. A higher accumulation of polyphenols was observed at higher doses of 24-EBL application (5.2 × 10^−7^ and 5.2 × 10^−8^ M), which indicates possible oxidative stress. All in all, this HPTLC method enabled a significant accumulation of polar and mid-polar polyphenols to be detected in the plumules and radicles of both corn seedling hybrids.

In addition to our findings from the analysis of the polyphenolic fraction of the antioxidant system of seedlings of two corn hybrids treated with 24-EBL, a survey is also provided of reports by many other researchers (Table 1) [23,25,26,27,28,32,34,37,38] who have worked on enzymatic and other non-enzymatic antioxidant systems of plants treated with BR phytohormones.

Data are also provided on the influence of BRs on seedling respiration, both overall for each of its components; this is important information not only for seedling energy, but also for antioxidant protection [23]. It has been observed that different BR (BL, 24-EBL, 28-HBL, CS, etc.) phytohormones, alone or in conjunction with other environmental factors (extreme temperatures, heavy metals, various protective and inhibitory substances, and osmotically active substances), act differently on various enzymatic (extreme temperatures, heavy metals, different protective and inhibitory substances, and osmotic active substances), acting differently in relation to enzymatic (SOD, POD, CAT, APX, DHAR, GR, GST, GPOX) and non-enzymatic (Pro, GSH, AA) systems in various plant species; the effect depends on the species, type of BR, and its concentration.

As the range of BR concentrations was mostly narrow in the considered research (Table 1), it is hard to say with certainty that antioxidant systems become activated only at high concentrations (of BR phytohormones), suggesting that additional tests are needed in this sense. An interesting finding was reported in the paper by Derevyanchuk et al. [23], which highlights the importance of COX and AOX respiration, i.e., cyanide-resistant and cyanide-sensitive respiration for the protection of plants from (salt) stress, showing evidently that such protection is enhanced due to BR treatment.

## 5. Changes in Thermodynamic and Energy Parameters During Seed Germination and the Early Stages of Growth and Development of Corn Seedlings—Implications for the Vigor of Young Corn Plants Influenced by BRs

The groups of data mentioned so far (see Figure 1a,b and Appendix A, and Table 1 and Appendix A) [21,22,23,25,26,27,28,29,30,31,32,33,34,37,38,39,44,45,46,55] regarding the influence of brassinosteroids on the early stages of plant growth and development also have an energy component. Since sugars, in addition to their building and signaling roles, are greatly important for plant energetics and seed germination, and the early stages of growth and development of plant seedlings are not supported by photosynthesis (see Appendix A) [22,29,30,31,33,35,37,38], we calculated the thermodynamic parameters for parts and whole seedlings of two corn hybrids (Appendix A) [21]. This was based on an idea related to different fractions of water (as a universal solvent in living matter): (1) free water in the extracellular space (symplast); (2) cytoplasmic water; and (3) water bound to biologically important molecules (e.g., proteins) by chemical forces have their own different energies [15].

Then, with correlation–regression calculations, we tested how much each of the monitored sugars contributed to the energy/thermodynamic changes in the seedlings of two corn hybrids (ZP434, ZP704). The model parameters of the regression relationship between the contents of selected sugars (Tre, Ara, Glu, Fru, Suc, Raf) and the differential thermodynamic parameters (entropy, enthalpy, and Gibbs free energy) assessed in 7-day-old corn seedlings (hybrids ZP434 and ZP704) are presented in Appendix A [21]. Gibbs free energy can be used to estimate the thermodynamic feasibility of a particular reaction or process in corn. A negative Gibbs free energy value indicates that a reaction is spontaneous and thermodynamically favorable, while a positive Gibbs free energy value suggests that the reaction is non-spontaneous and requires energy input. Measurements of Gibbs free energy changes during various biochemical reactions in corn can provide a better understanding of the metabolic pathways involved in processes such as photosynthesis, respiration, and carbohydrate metabolism. Appendix A and particularly Appendix A [21] clearly indicate that the regression between the content of sugars against differential entropy and enthalpy is inversely proportional, especially with respect to the monitored parameters in the plumulas and radicles of the seedlings of both corn genotypes. This relationship is very reminiscent of enthalpy–entropy compensatory dependence (Appendix A) [39], suggesting that the enthalpy–entropy relationship may influence the redistribution of assimilates in corn seedlings, and consequently changes in Gibbs free energy and seedling growth. The presented results are significant (Appendix A) [21] in terms of the possibility of maintaining the stability of these metastable structures in living matter via the controlled modulation of system entropy (by an inverse relationship between entropy and enthalpy changes) [17]. These results indicate that the changes in Gibbs free energy, particularly in seedling organs (plumule: P, radicle: R, rest of seed: RoS/S), occurring in both corn hybrids and their linear dependence on the monitored sugars are not statistically significant, in sharp contrast to the significant relationship observed in whole young plants (Appendix A) [21]. The observed correlations suggest that the regulation of these processes occurs at the plant seedling level. It indicates that different plant growth regulation processes (mainly regulated by different phytohormones) and/or source–sink relationships are important for the redistribution of assimilates, and therefore for the regulation of plant growth and the development of corn seedlings. This could be explained in more detail by examining the regression relationship between Gibbs free energy and the content of monitored sugars in whole young plants and individual organs (P, R, RoS/S).

The parameters of the PLS model for whole seedlings of theZP434 corn hybrid (involving 30 samples of parts of young plants: rest of seed + plumule + radicle: RoS/S + P + R) were statistically significant, with relatively high values of R^2^_cal_ and R^2^_CV_ and little difference between RMSEC and RMSECV values (Appendix A and Figure 2) [21]. The plot of the measured versus predicted ΔG^o^ values indicates the grouping of P and R samples on one side of the regression line and samples of RoS/S at the opposite end (Figure 2A) [21]. Additionally, in the group of RoS/S samples, the control samples are distinguished from the treated samples. Figure 2A is proof of the source–sink relationship between corn plumules and radicles, as young developing organs and net importers of assimilates (sink organs) and RoS/S, as a net exporter of assimilates (source organ) [56]. In the corn seedling ZP434 hybrid model, the medium values of exogenously added 24-EBLwere found to have the greatest influence on the source–sink ratio, which coincides with the results showing changes in maize seedling shoot length and other seedling growth parameters of various plant species/genotypes/ecotypes (Appendix A) [21,22,25,27,28,29,30,31,32,33,34,35,36,37,38].

The values of the ΔG^o^ parameter of thermodynamics in the plumule and radicle samples are evidently lower than in the RoS/S samples (Figure 2A), which indicates that the transfer of energy (and matter) occurs from the RoS/S to the plumule and radicle. This observation is understandable considering that the RoS/S is the source organ in corn seedlings, i.e., the origin of the net production of assimilates (mainly mono- and oligosaccharides) produced by the degradation of reserve starch, while the plumule and radicle are sink organs, i.e., they either consume or act as net “importers” of assimilates from the RoS/S [4,56].

Furthermore, the onset of grass seed germination is characterized by activated α-amylase enzyme synthesis, which is crucial in the process of starch degradation during grass seed germination. Again, this is controlled by the ratios between different phytohormones (GA and ABA) [4,6,56]. In addition, positive effects of the examined low-to-medium concentrations of 24-EBL on the growth of the seedlings of the ZP434 maize hybrid (Figure 1a and Appendix A) [39] can be assumed to be a possible consequence of the source–sink relationships in the corn seedlings driven by BRs (Figure 2A) [21]. Glucose has the highest positive impact on ΔG^o^ changes in whole ZP434 hybrid seedlings, followed by arabinose (Figure 2B) [21].

Glucose is a major substrate for mitochondrial respiration, and along with ATP production, it produces the organic acids necessary for transamination during nitrogen fixation [57,58]. Furthermore, glucose is principal monomer of cell wall polysaccharide [41], but also one of the constituents of sucrose, the major transport sugar in the vast majority of plants [59]. It is good to bear in mind that BR phytohormones affect some respiration patterns [60,61], but also the dynamics of cell wall construction [42], which is not surprising because the enzyme XET (enzyme which modifies plant cell walls) is induced (in addition to other factors resulting from the action of BRs [43]), which explains the observed relationship between the Glu and ΔG^o^ thermodynamic parameters.

A positive effect of arabinose (Figure 2B) on changes in the ΔG^o^ thermodynamic parameter would be similarly interpreted, since arabinose is one of the important monomers in arabinogalactan polymers of hemicellulose, an important building block of the cell wall [48].

The statistical parameters of the multilinear regression model for whole seedlings of the ZP704 corn hybrid are not the most adequate due to the relatively low values of R^2^_cal_ and R^2^_CV_, but the difference between the parameters RMSEC and RMSECV is not insignificant (Appendix A). Although the results for whole seedlings of the ZP704 corn hybrid (Figure 3A) [21] are more scattered relative to the equivalent data for whole seedlings of the ZP434 maize hybrid (Figure 2A) [21], it is evident (Figure 3A) [21] that the values of the thermodynamic ΔG^o^ parameter of this hybrid’s plumule and radicle are either lower or the same as the RoS/S of the ZP704 maize hybrid, a trend which is observed in whole seedlings of the ZP434 maize hybrid (Figure 2A) [21]. This could be associated with the source–sink relationship between plumules and radicles as sink organs (both young developing organs and net importers of assimilates) and the RoS/S as a source organ (net exporter of assimilates) [4,6,56].

Changes in the thermodynamic parameter ΔG^o^ (as a measure of the biosynthetic capacity of the system) obviously depend on the source–sink relationship, whereby ΔG^o^ “flows” from the RoS/S to the plumule and radicle (Figure 2A and Figure 3A) [21]. Regarding the ZP704 model, it was observed that a lower concentration of exogenously added 24-EBL had the greatest influence on the source–sink ratio, which is in accordance with the changes in maize seedling shoot length and other seedling growth parameters of various plant species/genotypes/ecotypes (Appendix A) [21,22,25,27,28,29,30,31,32,33,34,35,36,37,38]. Moreover, it has been noted that the source–sink transitions in young tissues are determined by the presence of sucrose synthase and invertase enzymes on target cells [62,63,64], which leads to the degradation of sucrose (a major transport sugar [59]) to its hexose components, and thus to the formation of a concentration gradient, which directs the phloem flow to the sink organs. Without going deeper into the question of whether 7-day-old corn seedlings have a developed phloem system or whether RoS/S sugars are transported to the plumule and radicle in a simpler manner, these processes, and thus the overall source–sink relationships, are highly influenced by plant hormones [64], affecting the sucrose-degrading enzymes.

The results (Appendix A, and Figure 2 and Figure 3) [21] indicate the importance of sugar redistribution in different organs (P, R, RoS/S) of corn seedlings and the effect of this on ΔG^o^ changes within a metabolic network. Activation depends on the genotype, the concentration of exogenously added 24-EBL, and the type of seedling organ. It has been mentioned that this diversity depends on the phenomenon of sink strength [56,64,65]. In addition, the effects of exogenously added 24-EBL on these processes in corn seedlings depend on the presence of BR plasmalemma receptors, i.e., so-called BRI proteins, which can vary in different tissues (e.g., in the root) [40]. Although the literature data strongly suggest that BR phytohormones are produced in plant meristem tissue [66], and since young plants are practically entirely composed of meristem tissue (additionally exposed to exogenous treatment with 24-EBL), we believe that the conducted experiment corresponds to the concentrations of BR-type phytohormones in a large number of plant tissues and species [66].

As we stated in our previous work [21], and as can be seen from the results (Figure 2A,B and Figure 3A,B, and Appendix A), we believe that the above experiments have confirmed the importance of sugar redistribution in different organs (P, R, RoS/S) of maize seedlings, as well as the influence of the aforementioned sugar redistribution in the organs of maize seedlings on changes in ΔG (Gibbs free energy) within the metabolic network of these seedlings, with their additional activation depending on the maize genotype, the concentration of exogenously added 24-EBL and the type of seedling organ. It was mentioned that this diversity depends on the phenomenon of sink strength [64,67]. In addition, the effects of exogenously added 24-EBL on these processes in maize seedlings depend on the presence of BR plasmalemma receptors (BRI proteins), which can vary in different tissues (e.g., the root) [40]. Although these phytohormones are thought to be produced in the plant meristem tissue [21], since young plants have a very pronounced meristem tissue (which was additionally exposed to exogenous treatment with 24-EBL), the described experiment corresponds to concentrations of BR-type phytohormones in a wide variety of plant tissues and species [68].

The canalization transport of auxins through the plant, which greatly influences various processes of plant growth and development [69,70,71] and related phenomena of apical dominance and phyllotaxis [71], is the reason why appropriate cybernetic solutions have been proposed [72]. However, other phytohormones have been reported to be transported through plants and to partially modulate the aforementioned effects of auxins [71]. BRs are among them [72]. In line with this, Hartwig and Wang [73] aimed to present a “molecular circuit of steroid signaling in plants”. Our contribution to all these problems consisted of taking into account both plant energetics and thermodynamics, resulting in the presented solution to the problem of source–sink relationships in plant seedlings. This indicates the possibility of quantitatively describing the process of plant seed germination and early seedling growth using thermodynamic parameters.

## 6. Conclusions and Further Directions Regarding the Mode of Action of BR Compounds

Although a quarter of a century has passed since the classic work of Clouse and Sasse [66], we believe that a physiological approach to examining the mode of action of BRs can provide an alternative and complementary insight into the mechanism of action of brassinosteroid phytohormones, compared to recent approaches in molecular biology. First of all, significant breakthroughs have been made over the past almost four decades in elucidating the role of brassinosteroids in coordinating various signaling processes and metabolic pathways at different physiological stages of plant growth and development [72,73,74,75,76,77], providing insight on their wider evolutionary context in the plant kingdom [66,73,78,79]. So, despite the context of these indisputable achievements in terms of the molecular approach to the problem of brassinosteroid action, the current methods require correction; it is necessary to point out several details and conclusions of this review reached by the classical physiological route, i.e., by analyzing the mode of action of brassinosteroids.

First, it is clear from the work of different researchers (Figure 1a,b, Figure 2, Figure 3, Appendix A, and Table 1, Appendix A) [19,20,21,23,25,26,27,28,29,30,31,32,33,34,35,36,37,53] that BRs, acting alone or in interaction with other compounds and environmental factors, function in specific ways on different organs of plant seedlings of various species. This conclusion is based on different concentrations of BRs that are optimal for causing one or another physiological or developmental effect in young plants of several species. It is particularly interesting that these effects occur even at very low BR concentrations (10^−12^–10^−15^ M of BRs), which cause a variety of effects on the growth and development of seedlings and young plants of different species. We also believe that this approach may provide more accurate results in some cases if, in addition to the usual monitoring of the temporal dynamics of brassinosteroid effects on various molecular and physiological processes, the influence of a wider range of concentrations of different BRs is also monitored simultaneously.

Secondly, the findings (Figure 2 and Figure 3, Appendix A) [21] provided by comparison of the thermodynamic parameter of Gibbs free energy and a group of some important sugars prove at least at the level of seedlings of young corn plants there is a transfer of matter (sugar) and energy (ΔG^o^) from the rest of the seed (source organ) to the plumule (shoot) and radicle (root), i.e., the sink organs. The whole process is exergonic, i.e., it takes place spontaneously. Time will tell if this or some other approach is most appropriate for proving the source–sink relationship in seedlings of other plants. Also, it is clear that BRs have a significant influence on these processes, but only at the level of whole seedlings of the two mentioned corn hybrids (ZP434, ZP704).

Last but not least, based on our previous hypothesis [80] and experiences and conclusions drawn by our team, as well as other groups of researchers who have practiced the classical physiological approach in researching the effects of brassinosteroids on the processes of germination and the early seedling growth of various plant species, we have some suggestions for further work that would be complementary to the molecular approach. Namely, instead of a destructive approach to calculating thermodynamic parameters (as applied in e.g., ref. [21]), measuring energetic changes in plants under the influence of BRs could be performed either by alternative methods of estimating thermodynamic parameters [81] or by IR thermography [82,83]. These approaches could measure or calculate the energetics of different BRs mutants and/or BRs genetically modified plants, thereby providing an alternative way to investigate brassinosteroid-dependent physiological phenomena in plants. The approach by groups of researchers from Ukraine and Belarus [23,60,61], who investigated the influence of BRs on various components of respiration of seedlings of various plants, seems to be promising and to produce interesting findings. Moreover, because the respiration of plant seedlings is part of the basic process of energy production, including the production of some important metabolites, e.g., organic acids needed for nitrogen metabolism in plants, is recommended [84,85].

Our findings and hypothesis are compatible with the findings of other researchers (Table 1), and we believe that using non-destructive methods for quantifying the energetics of seed germination and the early seedling growth of plants, especially mutants or GMO plants with altered pathways of brassinosteroid synthesis and signaling, together with measuring the sugar content and other energetically relevant plant compounds, can give us better insight into the mode of action of brassinosteroids at the whole-plant level. The already-established hypothesis of Sankar et al. [72] about brassinosteroids as modulators of auxin action at the whole-plant level is a good starting point for our proposed hypothesis.

## Figures and Tables

**Figure 1 ijms-26-02559-f001:**
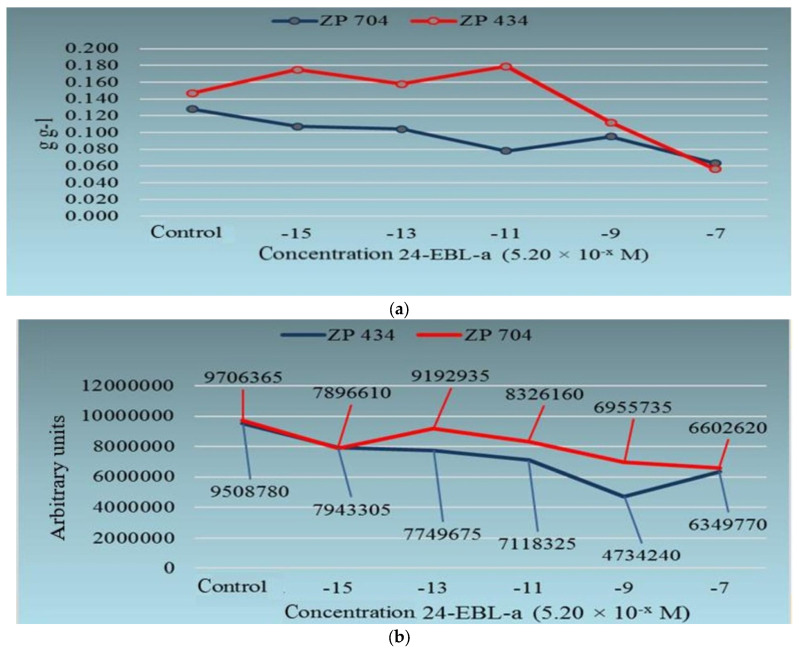
(**a**) Effects of different concentrations of 24-epibrassinolide (24-EBL) on vigor index II growth parameter (g g^−1^) of 7-day-old corn seedlings of ZP434 and ZP704 hybrids. Figure 1a is cited according to Waisi [40]. (**b**) Effects of different concentration of 24-EBL-a on content of ROS and RNS radical species in seedlings of the same corn hybrids. Figure 1b is cited according to Waisi [40].

**Figure 2 ijms-26-02559-f002:**
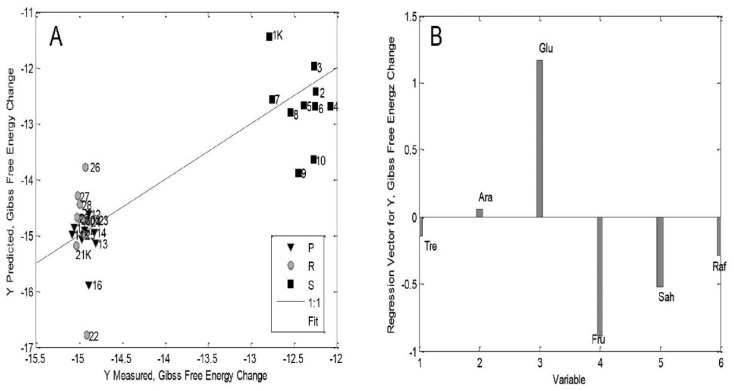
(**A**) The plot of the measured versus predicted ΔG^o^ values obtained from PLS model for whole 7-day-old corn seedlings of the ZP434 hybrid. “P” represents plumule, “R” represents radicle, and “S” represents rest of seed. K refers to control samples. Numbers from 1 to 30 represent concentrations of exogenously added 24-EBL to seedlings and seedling parts, from higher to lower concentrations, respectively. (**B**) Plot of the coefficients of descriptors (sugars: trehalose, arabinose, glucose, fructose, sucrose, raffinose) in PLS model of the whole 7-day-old corn seedlings of the ZP434 hybrid. Božilović et al. [21].

**Figure 3 ijms-26-02559-f003:**
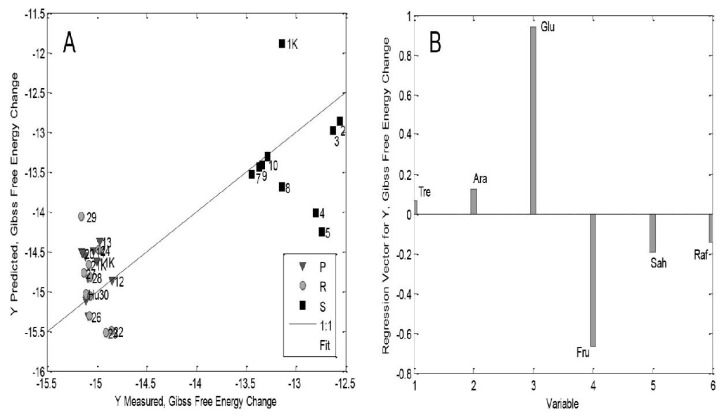
(**A**) The plot of the measured versus predicted ΔG^o^ values obtained from PLS model for whole 7-day-old corn seedlings of the ZP704 corn hybrid. “P” represents plumule, “R” represents radicle, and “S” represents rest of seed. K refers to control samples. Numbers from 1 K to 30 represent concentrations of exogenous 24-EBL added to seedlings from higher to lower concentrations, respectively. (**B**) Plot of the coefficients of descriptors (sugars: trehalose, arabinose, glucose, fructose, sucrose, raffinose) in the PLS model of whole 7-day-old corn seedlings of the ZP704 hybrid. Božilović et al. [21].

**Table 1 ijms-26-02559-t001:** Effect of different concentrations of various BRs with or without additional treatments on the content of different enzymatic and non-enzymatic antioxidative systems of plant species/genotypes/ecotypes. Abbreviations of enzymes and non-enzymatic antioxidant compounds: SOD: superoxide dismutase; POD: guaiacol peroxidase; APX: ascorbate peroxidase; GR: glutathione reductase; MDHAR: mono-dehydroascorbate reductase; DHAR: dehydroascorbate reductase; GST: glutathione-s-transferase; GPOX: glutathione peroxidase; Pro: proline, Glu: glutathione; AA: ascorbic acid; Tot resp.: total respiration; COX resp.: cyanid-resistant respiration; AOX resp.: cyanid-sensitive respiration. From various references.

Type of BR	Concentration of BRs	Other Treatments	Plant Species and/or Genotype/Ecotype; Age of Seedlings/Plants	Measured Parameters (with Unit of Measure)—9 Different Parts (S, R, St: Stem)	Higher (>, ≥)/Lower (˂, ≤) Values Than Control or Non-Significant Difference (n-s.d.)	Reference
24-EBL	10^−6^, 10^−8^ and 10^−10^ M	Ni (different concentrations)	*Vigna radiata* (nodulated) 45 d old	SOD, POD (units g^−1^ FW)	24-EBL treat.: ≥ Ni treat.: ≥ 24-EBL + Ni treat.: >	[38]
24-EBL	10^−6^, 10^−8^ and 10^−10^ M	Ni (different concentrations)	*Vigna radiata* (nodulated) 45 d old	CAT (mM H_2_O_2_ degraded g^−1^ FW), Pro (mol g^−1^ FW)	24-EBL treat.: n-s.d. Ni treat.: ≥ 24-EBL + Ni treat.: >	[38]
BRs	10^−5^ M	Different concentrations of Cd, GSH, GB, and SA	*Rice* (japonica type) 12 d after 2nd leaf stage	SOD (units g^−1^ FW),	(Cd, GSH, BRs, GB, SA treat.): > (S), n-s.d. (R, St)	[37]
BRs	10^−5^ M	Different concentrations of Cd, GSH, GB, and SA	*Rice* (japonica type) 12 d after 2nd leaf stage	POD (OD_470_ g^−1^ FW min^−1^)	(Cd, SA treat.): > (S); (GSH, BRs, GB treat.): n-s.d. (S); (GSH treat.): n-s.d. (R); (BRs treat.): n-s.d. (St); (BRs, GB, SA treat.): ˂ (R); (GSH, BRs, GB, SA treat.): ˂ (St);	[37]
Castasterone (CS)	10^−7^, 10^−9^ and 10^−11^ M	Cu (different concentrations)	*Brasicca juncea* 7 d old	APX, DHAR, GR, GST, GPOX (μmol UA mg^−1^ protein)	Controle + Cu: ≥ CS: n-s.d. CS + Cu: ≥	[34]
Castasterone (CS)	10^−7^, 10^−9^ and 10^−11^ M	Cu (different concentrations)	*Brasicca juncea* 7 d old	Glu, AA (mg g^−1^ FW)	Controle + Cu: ≥ CS: ≥ CS + Cu: ≥	[34]
28-HBL	10^−7^, 10^−9^ and 10^−11^ M	Zn (different concentrations)	*Brasicca juncea* 7 d old	POD, CAT, GR, APX, SOD (mmole UA mg protein^−1^)	Controle + Zn: >10^−7^ M (POD, GR, APX, SOD); Controle + Zn: >10^−9^ M (CAT) 28-HBL + Zn: >10^−7^ M (CAT, GR, APX, SOD)	[32]
28-HBL/ 24-EBL	0.5, 1.0, 2.0 10^−6^ M	PEG6000 (15% dilution)-induced drought	*Raphanus sativus* 3 + 7 d old	POD (μmol AA mg protein^−1^ min^−1^); CAT (μmol H_2_O_2_ mg protein^−1^ min^−1^)	Controle + PEG6000: > 24-EBL: > 2 × 10^−6^ M, >0.5 × 10^−6^ M PEG6000 + 24-EBL: >0.5 × 10^−6^ M 28-HBL: > 2 × 10^−6^ M, >0.5 × 10^−6^ M PEG6000 + 28-HBL: > 0.5 × 10^−6^ M	[27]
28-HBL/ 24-EBL	0.5, 1.0, 2.0 10^−6^ M	PEG6000 (15% dilution)-induced drought	*Raphanus sativus* 3 + 7 d old	APX (μmol AA mg protein^−1^ min^−1^); SOD (UA mg protein^−1^)	Controle + PEG6000: 24-EBL: >10^−6^ M, >0.5 × 10^−6^ M PEG6000 + 24-EBL: >0.5 × 10^−6^ M 28-HBL: >10^−6^ M, >0.5 × 10^−6^ M PEG6000 + 28-HBL: >0.5 × 10^−6^ M	[27]
24-EBL	10^−8^ M	H_2_O_2_ + cold (4 °C, 3 h during 3 d)	*Brasicca juncea* 10 d old	SOD (UA mg protein^−1^); CAT (UA mg protein^−1^); APX (μmol AA mg protein^−1^ min^−1^)	H_2_O_2_ + cold: n-s.d., >, > 24-EBL + H_2_O_2_ + cold: >	[26]
28-HBL/ 24-EBL	10^−6^, 10^−8^ and 10^−10^ M	Natural conditions	*Brasicca juncea* 10 d old	APX (μmol AA mg protein^−1^ min^−1^); CAT (UA mg protein^−1^)	>10^−6^ M 24-EBL and 28-HBL	[25]
24-EBL	10^−7^ M	Chlorpyrifos (0,06% dilution)	*Rice* (indica type) 12 d old	SOD (UA mg protein^−1^); GR (μmol min^−1^ mg protein^−1^); APX (μmol AA mg protein^−1^ min^−1^); CAT (UA mg protein^−1^); Pro (μmol g^−1^ FW)	SOD, GR, SOD: 24-EBL + CPF: >10^−7^ M; APX, Pro: 24-EBL +CPF: >10^−9^ M;	[30]
24-EBL	10^−6^, 10^−8^ and 10^−10^ M	Heat (40 °C, 3 h over 3 d)	*Brassica juncea* L. 10 d old	CAT (UA mg protein^−1^); APX (μmol AA mg protein^−1^ min^−1^); SOD (UA mg protein^−1^);	SOD, CAT: >10^−7^ M; APX: >10^−10^ M;	[28]
24-EBL	10^−8^ M	BZR, 50 mM and 100 mM of NaCl	*A.haliana* (ecotype Columbia 0) 21 d old	Tot, COX, and AOX respiration (resp.; nM CO_2_ min^−1^ X mg DW)	24-EBL + NaCl/ BZR Tot resp.: ≥/˂ COX resp.: ≥/≤ AOX resp.: ≥/˂	[23]

## Data Availability

The data presented in this study are available on request from the corresponding author.

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
