# Peer review of "Mode of Action of Brassinosteroids: Seed Germination and Seedling Growth and Development—One Hypothesis"

_ijms, 2025, doi:10.3390/ijms26062559_

Round 1

Reviewer 1 Report

Comments and Suggestions for Authors
  1. Lines 31-32: The terms "black box" and "gray box" are mentioned without detailed explanations, leaving the concepts unclear. Please explain these terms in detail.

  2. Line 77: Specify the selection criteria for the corn hybrids (ZP434 and ZP704) in the context of stress resistance.

  3. Lines 79-80: Describe the protocol for applying BRs (e.g., spraying, soaking) and justify the concentration ranges used.

  4. Lines 94-101: Integrate the germination parameters from Table S1 into the text.

  5. Lines 395-396: Discuss the practical implications of sugar redistribution in seedling organs (e.g., impact on productivity).

Suggestions

Include a comparison with similar studies on other species to establish the generality of the observed phenomena.

More discussion on recent advancements in molecular approaches to brassinosteroids could provide a balanced view and highlight how this study complements existing knowledge.

Author Response

For research article

Response to Reviewer X Comments

1. Summary

Thank you very much for taking the time to review this manuscript. Please find the detailed responses below and the corresponding revisions/corrections highlighted/in track changes in the re-submitted files. [This is only a recommended summary. Please feel free to adjust it. We do suggest maintaining a neutral tone and thanking the reviewers for their contribution although the comments may be negative or off-target. If you disagree with the reviewer's comments please include any concerns you may have in the letter to the Academic Editor.]

2. Questions for General Evaluation

Reviewer’s Evaluation

Response and Revisions

Does the introduction provide sufficient background and include all relevant references?

Yes/Can be improved/Must be improved/Not applicable

[Please give your response if necessary. Or you can also give your corresponding response in the point-by-point response letter. The same as below]

Are all the cited references relevant to the research?

Yes/Can be improved/Must be improved/Not applicable

Is the research design appropriate?

Yes/Can be improved/Must be improved/Not applicable

Are the methods adequately described?

Yes/Can be improved/Must be improved/Not applicable

Are the results clearly presented?

Yes/Can be improved/Must be improved/Not applicable

Are the conclusions supported by the results?

Yes/Can be improved/Must be improved/Not applicable

3. Point-by-point response to Comments and Suggestions for Authors

Comments 1: [Paste the full reviewer comment here.]

Response 1: [Type your response here and mark your revisions in red] Thank you for pointing this out. I/We agree with this comment. Therefore, I/we have.[Explain what change you have made. Mention exactly where in the revised manuscript this change can be found – page number, paragraph, and line.]

“[updated text in the manuscript if necessary]”

Comments 2: [Paste the full comment here.]

Response 2: Agree. I/We have, accordingly, done/revised/changed/modified…..to emphasize this point. Discuss the changes made, providing the necessary explanation/clarification. Mention exactly where in the revised manuscript this change can be found – page number, paragraph, and line.]

“[updated text in the manuscript if necessary]”

4. Response to Comments on the Quality of English Language

Point 1:

Response 1:    (in red)

5. Additional clarifications

[Here, mention any other clarifications you would like to provide to the journal editor/reviewer.]

For review article

Response to Reviewer 1 Comments

1. Summary

Thank you very much for taking the time to review this manuscript. Please find the detailed responses below and the corresponding revisions/corrections highlighted/in track changes in the re-submitted files. [This is only a recommended summary. Please feel free to adjust it. We do suggest maintaining a neutral tone and thanking the reviewers for their contribution although the comments may be negative or off-target. If you disagree with the reviewer's comments please include any concerns you may have in the letter to the Academic Editor.]

2. Questions for General Evaluation

Reviewer’s Evaluation

Response and Revisions

Is the work a significant contribution to the field?

We agree with intentions of reviewer No. 1

Is the work well organized and comprehensively described?

We agree with intentions of reviewer No. 1

Is the work scientifically sound and not misleading?

We agree with intentions of reviewer No. 1

Are there appropriate and adequate references to related and previous work? 

We agree with intentions of reviewer No. 1

Is the English used correct and readable?        

We agree with intentions of reviewer No. 1

3. Point-by-point response to Comments and Suggestions for Authors

Comments 1: Lines 31-32: The terms "black box" and "gray box" are mentioned without detailed explanations, leaving the concepts unclear. Please explain these terms in detail.

Response 1: Therefore, we clarified the cybernetic terms "black boxes" and "gray boxes" and have included them in the main body of the text, also with appropriate reference Ashby, 1956 (new reference No. 1). The changes are in lines 31-35 of the revised main text of the paper. The additions are in red,as per the IJMS journal rules.

Comments 2: Line 77: Specify the selection criteria for the corn hybrids (ZP434 and ZP704) in the context of stress resistance.

Response 2: Data on stress and drought resistance of the mentioned maize hybrids were taken from their manufacturer (Maize Research Institute, Belgrade, Serbia). These data are the result of several years of field trials conducted by researchers at that institute, which have been published hundreds of times in various scientific journals, both Serbian and international. After all, the mentioned two hybrids represent one of the most famous selections of the mentioned institute. Also, in the work of Waisi et al., 2015a (new reference No. 19), we examined the aforementioned two hybrids in field trials, and we also examined the differences in the chemical composition of the seeds of the mentioned two maize hybrids. The changes are in lines 81-83 of the revised main text of the paper. The additions are in red, as per the IJMS journal rules.

Comments 3: Lines 79-80: Describe the protocol for applying BRs (e.g., spraying, soaking) and justify the concentration ranges used.

Response 3: Seed treatment of two maize hybrids and their germination were performed using the classical ISTA method, which is described in detail as additional text in lines 85-95 of the revised text of the main part of the paper. The additions are in red, as per the IJMS journal rules.

Comments 4: Lines 94-101: Integrate the germination parameters from Table S1 into the text.

Response 4: Since the content of Tables S1 and S2 (from Supplementary data) is interconnected, we had provided a comment and explanation of their content in the corrected text located in lines 128-135 of the main part of the paper. In short, we compared the discriminativeness of the parameter % seed germination (Table S1) with other morpho-physiological parameters (Table S2) that describe seed germination and early stages of seedling growth of the two maize hybrids in question, as well as other plant species. Comparison of the discriminativeness of these two groups of morpho-physiological parameters (% germination and all other listed parameters) and concluded that it is first necessary to use such a wide range of 24-EBL (or any other brassinosteroid) concentrations in order to get a clear answer at which concentration of brassinosteroid the appropriate (positive or negative) response of plants occurs in terms of changing the given morpho-physiological parameter. The second conclusion is that the discriminativeness of the morpho-physiological parameters given in Table S2 is greater than that of the germination percentage, i.e. the parameter given in Table S1. This entire procedure is given as an introduction to further, more detailed analyses of the mode of action of various BRs phytohormones. The additions are in red, as per the IJMS journal rules.

Comments 5: Lines 395-396: Discuss the practical implications of sugar redistribution in seedling organs (e.g., impact on productivity).

Response 5: We disscused the practical implications of sugar redistribution in seedling organs in the corrected text located in lines 433-445 of the main part of the paper. The additions are in red, as per the IJMS journal rules.

4. Response to Comments on the Quality of English Language

Point 1: Reviewer No. 1 said that English language at the work was correct and readable

Response 1: We agree with English language oppinion of the reviewer No. 1

5. Additional clarifications: Suggestions of Reviewer No. 1: Include a comparison with similar studies on other species to establish the generality of the observed phenomena. More discussion on recent advancements in molecular approaches to brassinosteroids could provide a balanced view and highlight how this study complements existing knowledge.

Answer to Suggestions of Reviewer No. 1: Comparisons with the results of other researchers are given in detail in Table 1 of the main text of this paper, as well as in  Chapter 6. Conclusions and further research investigating the mode of action of BRs compounds, where it is stated which alternative, non-destructive methods for the assessment and/or measurement of energy parameters could be applied, as well as how these alternative methods, as well as the ones used here, could be combined with molecular methods for the examination of brassinosteroids, with other plant species, genotypes and mutants, including those with different brassinosteroid content, or BRI receptors for brassinosteroids. Relevant references and links to molecular methods for studying brassinosteroids are listed in the main text, as well as their links to physiological methods for studying the same phytohormones, so far, and it is suggested on our part how this could look in the future.

Reviewer 2 Report

Comments and Suggestions for Authors

Although the review deals with an interesting topic, there are several aspects that require major improvement.

The main points for improvement are as follows:

The review discusses the effects of brassinosteroids (BRs) on different plants (mainly corn hybrids) and highlights variances in response; it lacks a clear conclusion on the mechanisms leading to the diverse responses observed between hybrids under the influence of BRs. Authors should at least provide a hypothesis for the disparate effects or argue about this.

The review mentions that different concentrations of BRs can produce varying effects across organ systems ( , ), but it also highlights specific instances where very low concentrations (10^-12 to 10^-15 M) have significant physiological effects. There may be a need for further elaboration on how low concentrations can be as effective as higher ones, as this could imply a non-linear relationship that isn’t fully elucidated. Is there anything known on the receptors? Are there receptors able to respond at such low concentrations?

Although the review identifies significant correlations (e.g., energy transfer and sugar content), it does not deeply explore the biochemical pathways or mechanisms behind these correlations. This oversight might leave readers without a satisfactory understanding of how BRs exert their effects at the molecular level. Can you explain more of what is known or what is yet to be explored at the level of molecular pathways, given that the journal is called the International Journal of Molecular Sciences?

Figures 1a and 1b should be joined in the same figure, and please, change the scale in 1b in order to avoid large numbers on the Y axis. So many “0” are not informative.

Comments on the Quality of English Language

The English requires a thorough revision. The construction of many sentences is difficult to read and with grammar mistakes.  

Line 424: "on"

Author Response

For research article

Response to Reviewer 2 Comments

1. Summary

Thank you very much for taking the time to review this manuscript. Please find the detailed responses below and the corresponding revisions/corrections highlighted/in track changes in the re-submitted files. [This is only a recommended summary. Please feel free to adjust it. We do suggest maintaining a neutral tone and thanking the reviewers for their contribution although the comments may be negative or off-target. If you disagree with the reviewer's comments please include any concerns you may have in the letter to the Academic Editor.]

2. Questions for General Evaluation

Reviewer’s Evaluation

Response and Revisions

Does the introduction provide sufficient background and include all relevant references?

Yes/Can be improved/Must be improved/Not applicable

[Please give your response if necessary. Or you can also give your corresponding response in the point-by-point response letter. The same as below]

Are all the cited references relevant to the research?

Yes/Can be improved/Must be improved/Not applicable

Is the research design appropriate?

Yes/Can be improved/Must be improved/Not applicable

Are the methods adequately described?

Yes/Can be improved/Must be improved/Not applicable

Are the results clearly presented?

Yes/Can be improved/Must be improved/Not applicable

Are the conclusions supported by the results?

Yes/Can be improved/Must be improved/Not applicable

3. Point-by-point response to Comments and Suggestions for Authors

Comments 1: [Paste the full reviewer comment here.]

Response 1: [Type your response here and mark your revisions in red] Thank you for pointing this out. I/We agree with this comment. Therefore, I/we have.[Explain what change you have made. Mention exactly where in the revised manuscript this change can be found – page number, paragraph, and line.]

“[updated text in the manuscript if necessary]”

Comments 2: [Paste the full comment here.]

Response 2: Agree. I/We have, accordingly, done/revised/changed/modified…..to emphasize this point. Discuss the changes made, providing the necessary explanation/clarification. Mention exactly where in the revised manuscript this change can be found – page number, paragraph, and line.]

“[updated text in the manuscript if necessary]”

4. Response to Comments on the Quality of English Language

Point 1:

Response 1:    (in red)

5. Additional clarifications

[Here, mention any other clarifications you would like to provide to the journal editor/reviewer.]

For review article

Response to Reviewer 2 Comments

1. Summary

Thank you very much for taking the time to review this manuscript. Please find the detailed responses below and the corresponding revisions/corrections highlighted/in track changes in the re-submitted files. [This is only a recommended summary. Please feel free to adjust it. We do suggest maintaining a neutral tone and thanking the reviewers for their contribution although the comments may be negative or off-target. If you disagree with the reviewer's comments please include any concerns you may have in the letter to the Academic Editor.]

2. Questions for General Evaluation

Reviewer’s Evaluation

Response and Revisions

Is the work a significant contribution to the field?

[We have some onjections on those opinion of Reviewer No. 2 on the work. Our answers located below.

Is the work well organized and comprehensively described?

[We have some onjections on those opinion of Reviewer No. 2 on the work. Our answers located below.

Is the work scientifically sound and not misleading?

[We have some onjections on those opinion of Reviewer No. 2 on the work. Our answers located below.

Are there appropriate and adequate references to related and previous work? 

[We have some onjections on those opinion of Reviewer No. 2 on the work. Our answers located below.

Is the English used correct and readable?        

- - - -

We have some onjections on those opinion of Reviewer No. 2 on the work. Our answers located below.

3. Point-by-point response to Comments and Suggestions for Authors

Comments 1: The review discusses the effects of brassinosteroids (BRs) on different plants (mainly corn hybrids) and highlights variances in response; it lacks a clear conclusion on the mechanisms leading to the diverse responses observed between hybrids under the influence of BRs. Authors should at least provide a hypothesis for the disparate effects or argue about this.

Response 1: We realy glad for the criticality of reviewer No. 2, but we would first point out the title of the paper, which reads "Mode of action of brassinosteroids. ...", which indicates the secondary effects of brassinosteroids in plants in the early stages of germination and development of seedlings of various plant species, including the used corn hybrids. Therefore, our conclusion about the action of brassinosteroids is necessarily indirect, but not devoid of a molecular approach. After all, isn't thermodynamics, which is used in the analysis of the effects of brassinosteroids, a type of analysis of molecular processes, expressed through statistical relations!? At least since Boltzmann in physics, and since Ilya Prigogine in biology, it has been a legitimate way of analyzing complex molecular processes, which can only be treated statistically. Therefore, our hypotheses are given very cautiously, and as a PROPOSAL for further analysis, which could be carried out by colleagues who deal with the molecular aspects of the action of brassinosteroids. Anyway, thanks to the suggestions of reviewer No. 1, a working hypothesis is given at the end of chapter 5 of those article, although it is conceived more as a question, but the limitations of the methods that I and my collaborators used, as well as the methods of the works that we cited, necessarily impose a cautious approach in drawing conclusions about such complex processes. Also, in the final chapter 6, we referred to non-destructive methods of analyzing plant energetics, for example: measuring respiration, as conducted by Ukrainian and Belarusian researchers (see ref. Derevyanchuk et al., 2014; Pokotylo et al., 2014; Derevyanchuk et al., 2017 and Derevyanchuk et al., 2019, cited in ref. 19 (Božilović et al, 2023) and. ref. 73 (Waisi et al., 2019), as well as the application of thermography, in such early stages of their development, as a way to non-destructively quantify the energetics of various genotypes (produced by selection or mutation) of various plant species. We propose that such an approach to non-destructive quantification of the energetics of various genotypes or plant species can be applied in parallel with the quantification of brassinosteroid levels in these genotypes and plant species, then by quantifying the levels of BRI plasmalemmal receptors for these phytohormones, as well as possibly by quantifying the content of enzymes that transform sugars (invertase, sucrose synthase, etc.) in the early stages of development of both maize and other plants. In this way, the already mentioned cybernetic approach of the "black", i.e. "gray box" analysis of "inputs" and "outputs" from the analyzed plant systems would be combined with the molecular approaches that we mentioned. Ultimately, the main intention of this review is not only to resolve some issues, but also to view these same issues from a DIFFERENT ANGLE, thereby contributing to the elucidation of the very complex effect of brassinosteroids on plant growth and development, especially in the very early stages, which we call the general name seedling establishment, and one of the key stages for the bioproductivity of native and crop plants. A purely cybernetic hypothesis about the action of brassinosteroids through the regulation of the flow of auxin phytohormones in plants is discussed in the work of Sankar et al. 2011 (see ref. 19 (Božilović et al, 2023) and. ref. 73 (Waisi et al., 2019), but we believe that a cybernetic consideration of the relationship between brassinosteroids, energetics and the redistribution of various sugars in parts of seedlings, here of corn, somewhere else of other plant species, can be at least an interesting, if not fruitful, approach.

Comments 2: The review mentions that different concentrations of BRs can produce varying effects across organ systems ( , ), but it also highlights specific instances where very low concentrations (10^-12 to 10^-15 M) have significant physiological effects. There may be a need for further elaboration on how low concentrations can be as effective as higher ones, as this could imply a non-linear relationship that isn’t fully elucidated. Is there anything known on the receptors? Are there receptors able to respond at such low concentrations?

Response 2: A very good question from the referent No. 2, which opens an interesting and still insufficiently clear area of the homeopathic effect of various substances on the physiological responses of plants. So far, we have only concluded that the effect of the entire range of 24-EBL concentrations used is nonlinear, that it also affects not only the energetics and redistribution of various elements and sugars in the organs of corn seedlings (see Tables and Graphs in the main text of the paper and Supplementary Data), but also affects the physiological responses not only of the tested corn hybrids, but also of a whole range of other plant species (see Table 1 in the main part of the paper). So far, we are only aware of the work of Van Esse et al., 2012 (see in ref. 19: Božilović et al., 2023), which discusses the issue of unequal redistribution of BRI plasmalemmal receptors for brassinosteroids in the roots of Arabidopsis thaliana, so we would not dare, without insight into the works of experts in this field of research on these phytohormones, to draw any far-reaching conclusions. We think that this is an interesting hypothesis, BUT, , since we do not deal with the molecular aspects of the action of brassinosteroids, we leave the solution of this problem to colleagues who do. Our goal was, I repeat, to shed light on this problem from a DIFFERENT ANGLE.

Comments 3: Although the review identifies significant correlations (e.g., energy transfer and sugar content), it does not deeply explore the biochemical pathways or mechanisms behind these correlations. This oversight might leave readers without a satisfactory understanding of how BRs exert their effects at the molecular level. Can you explain more of what is known or what is yet to be explored at the level of molecular pathways, given that the journal is called the International Journal of Molecular Sciences?

Response 3: The field of thermodynamics is the exploration of molecular processes in various systems, including biological ones, through a statistical approach. We believe that we have proven the connection between changes in energy and thermodynamic parameters and the redistribution of sugars and various elements in various organs of maize seedlings, and we have proposed, based on the works of other researchers, who also worked in a wider range of brassinosteroid concentrations (see Table 1 in the main part of the text), that instead of the destructive approach in determining thermodynamic and energy parameters (which we used in our previous works and whose results we discuss here in a broader context), research on the same phenomena should be conducted using non-destructive methods of quantifying energy (respiration measurement and thermography) of plants in the early stages of plant growth and development (i.e. during seed germination and seedling establishment phases), and all this combined with other physiological and molecular methods of researching the effects of brassinosteroids. Once again, this is a SECOND ANGLE of research into the effects of brassinosteroids on the aforementioned phases of plant growth and development.

Comments 4: Figures 1a and 1b should be joined in the same figure, and please, change the scale in 1b in order to avoid large numbers on the Y axis. So many “0” are not informative.

Response 4: In lines 124-129 (original text of the main part of the paper) we wrote the following: "Figures 1a and 1b indicate a similarity in the effect of different concentrations of exogenous 24-EBL on the seedling growth parameter vigor index II and their influence on changes of the corn seedling content of ROS and RNS (oxygen and nitrogen) radical species, which are significant in numerous developmental processes and plant growth, such as remodeling of plant cell wall [39, 40], cell division and growth of young plants [41]." In short, we have established the EXTERNAL similarity of two different processes, which should serve as an introduction to the further discussion. So it is understandable that these two graphs, although connected by the external SIMILARITY of different processes, must nevertheless, since this is an introduction, be separated. Therefore, the differences in the numerical values of the ordinates cannot be the same, simply because they describe different processes. Our goal was to point out through this introduction the possible similarity and connection of two otherwise different processes, i.e. processes MAYBE of the same biological nature, but which take place at different levels of biological structures and functions. After all, the problem of seed germination and seedling establisment is of crucial importance from both a biological and an economic point of view and can be studied at an extremely reduced level by measuring the degree of germination, but also by very sophisticated methods, such as determining the content of ROS and RNS radicals in corn seedlings. Our goal was to move from simple to more complex insights. The methods we used are, we admit, of a limited level of knowledge, BUT they indicate a connection between the action of brassinosteroids and a series of physiological responses, obtained both by our research team and by a number of researchers from many countries (see Table 1, main part of the text of the paper), whereby we particularly emphasized the connection between the energetics of seed germination and seedling establishment of seedlings of two maize hybrids with other physiological processes. We did not deal with the molecular aspects of these processes, we only point out that the energetics of these processes from the point of view of researching the action of brassinosteroids has not been sufficiently explored.

4. Response to Comments on the Quality of English Language

Point 1: The English requires a thorough revision. The construction of many sentences is difficult to read and with grammar mistakes.

Response 1: Dear Reviewer No. 2, the English text of this paper is a slightly revised version (less than 5% of the total text) of an earlier version, which underwent MDPI language revision. Secondly, before that, this text was reviewed by a person who has over 25 years of experience in translating works in the field of biology and biotechnology into English, so that the text of this paper, which is the work of the above-listed authors, and primarily the corresponding author, was essentially double-checked, from other peoples, including MDPI language revision staff. We can send material proof of this to the people at IJMS magazine, MDPI publisher.

5. Additional clarifications

Other comments, changes in the main text of the paper, etc. were made based on the suggestions of reviewer no. 1, including two additional references (in the new version of the text, references no. 1 and 19).

Round 2

Reviewer 2 Report

Comments and Suggestions for Authors

Most of my comments have not been considered. Figures 1a and b are not present in this version. Most of my doubts are still present in the current version, so I cannot recommend publication.

Author Response

Remaining comments from reviewer 2 and and the responses of the first author of the paper IJMS-3431492

Comment 1: The review discusses the effects of brassinosteroids (BRs) on different plants (mainly corn hybrids) and highlights variances in response; it lacks a clear conclusion on the mechanisms leading to the diverse responses observed between hybrids under the influence of BRs. Authors should at least provide a hypothesis for the disparate effects or argue about this.

  • Although your comments surrounding the focus of your article to be on energetics and not wanting to draw conclusions into work you are not familiar with, the reviewer asks a hypothesis and not a conclusion

Comments 2: The review mentions that different concentrations of BRs can produce varying effects across organ systems ( , ), but it also highlights specific instances where very low concentrations (10^-12 to 10^-15 M) have significant physiological effects. There may be a need for further elaboration on how low concentrations can be as effective as higher ones, as this could imply a non-linear relationship that isn’t fully elucidated. Is there anything known on the receptors? Are there receptors able to respond at such low concentrations?

Answers to Comment 1 and Comment 2: The end of Chapter 5 has been revised in the Completeness, by removing the previous paragraph and adding two new paragraphs (marked in red), as a quote from reference (21). Thus, several possible scenarios of the influence of altered concentrations of brassinosteroids, which ultimately lead to altered redistribution of carbohydrates and energy parameters in the organs of seedlings of the two maize genotypes, are listed. Also, newly introduced references in the Literature are marked in red.

Comments 3: Although the review identifies significant correlations (e.g., energy transfer and sugar content), it does not deeply explore the biochemical pathways or mechanisms behind these correlations. This oversight might leave readers without a satisfactory understanding of how BRs exert their effects at the molecular level. Can you explain more of what is known or what is yet to be explored at the level of molecular pathways, given that the journal is called the International Journal of Molecular Sciences?

-where you have proposed non-destructive methods combined with other physiological and molecular methods- this could mention the molecular methods intended or the mechanisms it would be beneficial or complementary to you work to elucidate in the future

-similarly, a brief mention of anything known about the receptors in question, or rather, what is not know would suffice

Answers to Comment 3: As mentioned in my answer to questions 1 and 2, several possible scenarios of molecular processes, triggered by altered concentrations of brassinosteroids, were listed, which ultimately lead to changes in the redistribution of carbohydrates and energy parameters in the kernels of two maize seedlings. Furthermore, in the previous answer to reference no. 2, I stated that since the time of Ludwig Boltzmann in physics and Ilya Prigogine in biology, thermodynamic methods have been a legitimate method for investigating molecular processes in living systems through statistical thermodynamics, as we have demonstrated in this review. We also mentioned in the sentence "Namely, instead of a destructive approach in calculation of thermodynamic parameters, should be done either via alternative estimates of thermodynamic parameters [81] or via IR thermography [82; 83] in order to differentiate the energetics of different mutants an/or genetically modified plants for the synthesis/metabolism/signaling of BR (single action or in „cooperation“ with other physiologically active substances) in plants." (lines 506-511 in the revised text we are sending, i.e. in lines 488-493 in the previous version of the text) that the mentioned physiological approach, destructive as well as non-destructive (which was used by other researchers) can be used for testing mutants or genetically modified plants with altered synthesis/metabolism of the signaling function of brassinosteroids. We believe that this more than adequately responded to the requests of reviewer no. 2.

Comment 4: Figures 1a and 1b should be joined in the same figure, and please, change the scale in 1b in order to avoid large numbers on the Y axis. So many “0” are not informative.

  • The Y-axis scale could be changed to reduce the amount of ‘0’’s

Answers to Comment 4: In the case of Figures 1a and 1b, it is stated that they are taken as quotes from the work of Waisi (39). Any CHANGES to that figure, either by merging them or changing the numbering of the axes on them, is a COPYRIGHT VIOLATION, which reviewer no. 2 knows very well. Since we do not want to be accused of plagiarism by the author of the work of Waisi (39), but only want to stick to the citation, we refuse to make changes to the aforementioned Figures 1a and 1b.
